# Expansion and Functional Diversification of *SKP1-Like* Genes in Wheat (*Triticum aestivum* L.)

**DOI:** 10.3390/ijms20133295

**Published:** 2019-07-04

**Authors:** Imen HajSalah El Beji, Said Mouzeyar, Mohammed-Fouad Bouzidi, Jane Roche

**Affiliations:** UMR INRA/UCA 1095 GDEC, Université Clermont Auvergne, Campus Universitaire des Cézeaux, 1 Impasse Amélie Murat, TSA 60026 AUBIERE, France

**Keywords:** SKP1, *Triticum aestivum*, structural and evolution analysis, Y2H

## Abstract

The ubiquitin proteasome 26S system (UPS), involving monomeric and multimeric E3 ligases is one of the most important signaling pathways in many organisms, including plants. The SCF (SKP1/Cullin/F-box) multimeric complex is particularly involved in response to development and stress signaling. The SKP1 protein (S-phase kinase-associated protein 1) is the core subunit of this complex. In this work, we firstly identified 92 and 87 non-redundant *Triticum aestivum*
*SKP1-like* (*TaSKP*) genes that were retrieved from the latest release of the wheat genome database (International Wheat Genome Sequencing Consortium (IWGSC) RefSeq v1.0) and the genome annotation of the TGAC v1 respectively. We then investigated the structure, phylogeny, duplication events and expression patterns of the *SKP1-like* gene family in various tissues and environmental conditions using a wheat expression platform containing public data. *TaSKP1-like* genes were expressed differentially in response to stress conditions, displaying large genomic variations or short insertions/deletions which suggests functional specialization within *TaSKP1-like* genes. Finally, interactions between selected wheat FBX (F-box) proteins and putative ancestral TaSKP1-like proteins were tested using the yeast two-hybrid (Y2H) system to examine the molecular interactions. These observations suggested that six *Ta-SKP1* genes are likely to be ancestral genes, having similar functions as *ASK1* and *ASK2* in Arabidopsis, *OSK1* and *OSK20* in rice and *PpSKP1* and *PpSKP2* in *Physcomitrella patens*.

## 1. Introduction

Selective degradation of proteins plays a key role in diverse aspects of eukaryotic mechanisms such as responses to physiological development (flowering, cell cycle, etc.) or in responses to diverse environmental stresses [1,2,3,4,5,6] by degrading target proteins that play a role in the activation or repression of downstream genes. The substrate selectivity is insured by E3 ubiquitin ligases which could be mono- or multi-subunit complexes. Among the multi-subunit E3 ubiquitin ligases complexes, the SKP1/Cullin 1/F-box protein (SCF) family is particularly well characterized [7,8]. Each SCF complex is composed of four protein components: the Cul1-Rbx1 (Cullin 1/ RING-box protein 1) catalytic core bound to a variable F-box protein (FBX)-Skp1 substrate recognition module [9,10].

SKP1 is a small protein of approximately 160 amino acids and functions as a core component connecting CUL1 and an F-box protein. SKP1 plays crucial roles in cell-cycle progression, in hormone and light signaling [11,12,13,14,15,16] as well as in vegetative and flower development [17,18,19,20,21].

Only one *SKP1* gene has been described in protists, algae and fungi [22]. In contrast, plants have multiple *SKP1* genes that appeared as a consequence of duplication events leading to diverse functions [22,23,24,25,26]. *SKP1-like* gene evolution in several plant species was investigated by Kong et al. [22] who suggested that all the *SKP1* genes found in these species were originated from one ancestral gene called *ASK1* in Arabidopsis and *OSK1* in rice, which may have similar functions [22,26]. Further, *SKP1* genes were described as belonging to three different classes by Kong et al. [26]. Type Ia corresponds to *SKP* genes containing one intron, Type Ib corresponds to intronless *SKP* genes, and Type II corresponds to *SKP* genes containing more than one intron. Furthermore, Kong et al. [26], then Kahloul et al. [23], suggested that ancestral *SKP1* genes typically contain one intron whose position is conserved within the genomic sequence. 

To gain further insight into the characterization of wheat *SKP1-like* genes, we carried out a comprehensive analysis of this gene family. We investigated the structure, phylogeny, duplication events and expression patterns of the *SKP1-like* gene family in various tissues and environmental conditions using a wheat expression platform [27]. All *Triticum aestivum SKP1-like* (*TaSKP1*) genes were retrieved using data from the latest release of the wheat genome database (International Wheat Genome Sequencing Consortium (IWGSC) RefSeq v1.0, https://www.wheatgenome.org/) and TGAC v1 [28] annotated genome reference. Analysis of the structure, the expansion and the expression of this family of genes was conducted. In addition, we also analyzed the homoeology relationships between the subgenomes A, B and D that constitute the hexaploid wheat genome. Because SKP1 and FBX proteins interact to form the SCF complex, the yeast two-hybrid (Y2H) system was used to test for the physical interactions between certain wheat TaSKP1 proteins, considered to be ancestral proteins, and three wheat FBX already cloned in the laboratory (TaFBX proteins).

## 2. Results

### 2.1. Genome-Wide Identification of TaSKP1-Like Genes

HMMER2-BLASTP-InterProScan searches of databases containing the latest version of the annotated wheat genome database (International Wheat Genome Sequencing Consortium (IWGSC RefSeq v1.0))and in TGAC v1 [28] identified 92 (which we designated *TaSKP1_1* through *TaSKP1_92*) and 87 non-redundant *Triticum aestivum SKP1-like* genes, respectively, with conserved Skp1 and Skp1_POZ (POxvirus and Zinc finger) domains.

Subsequently, we established the correspondence (one-to-one) between the two annotations for the *TaSKP1-like* genes (Appendix A).

### 2.2. Chromosomal Location of TaSKP1-Like Gene

By using their coordinates on the pseudomolecule, we found that the 92 sequences were located on three subgenomes A, B and D (Table 1). It appears that each chromosome of each subgenome contains at least one *TaSKP1-like* gene (Figure 1). However, the distribution between chromosomes was not even. Only chromosomes Chr1A and Chr1B each contained a single copy, whereas the other chromosomes contained several *TaSKP1-like* genes. For example, the 3D, 3B and 5D chromosomes contained 8, 9 and 10 copies, respectively. In addition, when a chromosome contains several copies, some of these copies are grouped into clusters and sometimes in a very narrow region such as the *TaSKP1_49, TaSKP1_50, TaSKP1_51* and *TaSKP1_52* genes, which were clustered within 61 kb on the 2D chromosome or the *TaSKP1_57* and *TaSKP1_58* genes which were clustered within 72 kb on the 3B chromosome. In addition, *TaSKP1-like* genes belonging to the same family tended to be clustered within the same chromosomal location. For instance, *TaSKP1_47* through *TaSKP1_53*, which are intronless genes, were clustered in the distal part of the 2D chromosome. Similarly, *TaSKP1_23*, *TaSKP1_24, TaSKP1_25* and *TaSKP1_26,* which contain one intron in a conserved position, were located within the same region on the 7B chromosome (Figure 1).

### 2.3. Phylogeny of the TaSKP1-Like Genes

As shown in Figure 2, a phylogenetic tree of wheat *TaSKP1-like* genes was inferred using the maximum likelihood method. Several clades contained genes belonging to the same class determined by Kong et al. [26] and Kahloul et al. [23]. In particular, it can be noticed that there was a large clade comprising 23 *TaSKP1-like* genes, among which 22 were intronless (class Ib genes) and only one gene (*TaSKP1_83*) contained several introns (class II). Moreover, the length of the branches in this clade reflects a high sequence similarity between its members. Similarly, one clade included most of the genes with more than one intron (*TaSKP1_86, TaSKP1_82, TaSKP1_87, TaSKP1_84, TaSKP1_88, TaSKP1_85* and *TaSKP1_89*) (class II). 

The grouping of some sequences with the same structure on the phylogenetic tree as well as the clustering of the same genes on the wheat pseudomolecule suggest that they could be the result of duplication phenomena.

### 2.4. Homoeologous Group Identification, Structure and Expression Profiles of TaSKP1-Like Genes

To identify homoeologous sequences, CD-HIT suite was used with sequence identity cut-off set at 90%, meaning that sequences with identity percent above 90 are clustered together. So, 42 clusters were identified and numbered from cluster #0 to cluster #41 (Appendix A). The cluster #0 was the largest one and contained 18 sequences. Twenty-three clusters (clusters #19 through #41) each contained only one sequence. To identify homoeologous sequences, clusters containing two or three members that were originating from the same chromosome in subgenomes A, B and D were considered as homoeologous groups. For example, cluster #3 contained TaSKP1_54, TaSKP1_58 and TaSKP1_61, indicating that it is likely a triplet of homoeologous genes in the third chromosome. Similarly, cluster #13 contained TaSKP1_44 and TaSKP_53 corresponding to a pair of homoeologous genes. In contrast, cluster #11 and cluster #12 contained pairs of highly similar sequences but were located in non-homoeologous chromosomes. In addition, cluster #0 and cluster #2 contained more than three members, among which some were considered as “probable homoeologs” because they were located in homoeologous chromosomes (Appendix A).

In summary, eight triplets and 11 pairs yielding a total of 46 members were identified as homoeologous groups, while the 46 remaining sequences were considered as “singlets” without any obvious homoeologous counterpart.

Considering the homoeolog structure, all the homoeologs displayed the same exon-intron organization, except cluster #7 and cluster #0. In cluster #7, TaSKP1_34 contained one intron while TaSKP1_67 and TaSKP1_72 did not, suggesting that the A copy could have acquired an intron after polyploidisation. Simarly, TaSKP1_76 contained one intron while its probable homoeolog TaSKP1_77 was intronless.

Then, homoeolog triplets’ and pairs’ expression profiles were followed in different seed tissues (grains, spikes, leaves and roots) [29]. Independently to the homoeologous groups identified, TaSKP1-like genes expression patterns appear to be diverse (Appendix A). In the case of triplets, five of them exhibited similar expression between the homoeologs, while three triplets experienced partitionning of expression between the homoeologs. Indeed, one or two homoeologs were expressed differently when compared to the other homoeologs. For example, TaSKP1_7, TaSKP1_10 and TaSKP1_14 formed a triplet within which, the A and B homoeologs were more expressed thant the D homoeolog. The same situation was observed concerning pairs of homoeologs. For example, while TaSKP1_4 (B subgenome) and TaSKP_5 (D subgenome) exhibited almost identical expression profiles, the pair formed by TaSKP1_56 (B subgenome) and TaSKP1_60 (D subgenome) exhibited a very contrasting expression with no expression detected for the D homoeolog.

### 2.5. Duplications

In order to investigate the hypothesis of duplication, approximative lengths of 100 kb on either side (i.e., regions of about 200 kb), of each *TaSKP1-like* gene belonging to the same clade was selected for further investigation. As an example, a clade containing 23 intronless genes was selected as indicated in yellow in Appendix A and Appendix A. In the cases of *TaSKP1_49, TaSKP1_50, TaSKP1_51* and *TaSKP1_52*, the genes were very close (separated by less than 100 kb), hence a region of 261.018 kb containing these genes was used. Similarly, a region of 255.143 kb which contained *TaSKP1_74* and *TaSKP1_75* was used. Using the Yass tool, several highly conserved regions were identified between these two regions, which are likely the result of duplication phenomena. This is illustrated in Figure 3, which represents duplicated fragments that can be identified. To clarify the nature and extent of duplication, we identified all duplicated fragments between loci that were at least 3 kb in size and had 90% identity (Appendix A). Thus, in total, we identified 122 non-overlapping duplicate fragments whose lengths ranged from 3.098 to 12.619 Kb (Figure 4a and Appendix A). Figure 4b presents duplication relationships between members of these clades. The largest number of duplicate fragments between two non-homoeologous loci involved the *TaSKP1_49*-*TaSKP1_50*-*TaSKP1_51*-*TaSKP1_52* locus on the 2D chromosome which shared 16 fragments with other loci, and the *TaSKP1_74*-*TaSKP1_75* locus on the 5D chromosome which shared 29 fragments with other loci. The majority of duplications were around 4 kb (median duplication size = 5.960 kb, mean duplication size = 6.208 Kb). The largest duplicate fragment was 13,675 kb in length between the loci containing *TaSKP1_74* and *TaSKP1_75* on the 5D chromosome and the locus containing *TaSKP1_47* on the 2D chromosome.

### 2.6. Structural and Phylogenic Analysis of TaSKP1-Like Genes

Nucleotide and protein alignments indicated that *TaSKP1_1* to *TaSKP1_27* genes contained a unique intron in a conserved position, meaning that the second exon is 174 bp long while the first exon and the intron could be variable in length. These genes were all classified as Type Ia *SKP* genes under Kong et al.’s system. *TaSKP1_28* to *TaSKP1_41* genes contained one non-conserved intron and were all classified as Type Ia. *TaSKP1_42* to *TaSKP1_81* intronless genes were all classified as Type Ib. *TaSKP1_82* to *TaSKP1_92* genes contained more than one intron and were all classified as Type II.

Phylogenetic analysis of the 174 *SKP1-like* genes from the moss, monocot and eudicot species (see Material and Methods) indicated that *TaSKP1_3, TaSKP1_15, TaSKP1_16, TaSKP1_19, TaSKP1_21, TaSKP1_27* and six other monocot *SKP1-like* genes (*OSK1, ZmSKP1, SbSKP1, SbSKP2, BdSKP1* and *BdSKP2*) that contained a single position-conserved intron clustered together (Figure 2). In contrast, the remaining genes clustered in lineage-specific clades which suggests that they formed after the split between wheat and the other species. Interestingly, while the majority of wheat *TaSKP1-like* genes clustered together, some of them were found in clades containing genes originating from other monocots. For instance, *TaSKP1_3, TaSKP1_15, TaSKP1_16, TaSKP1_19, TaSKP1_21* and *Ta SKP1_27* were found in a clade containing *ZmSKP1* from maize, *OSK1* from rice, *BdSKP1* and *BdSKP2* from *Brachypodium* and *SbSKP2* from sorghum. Indeed, all these genes contain a single intron in a conserved position. A second clade also contained mainly *TaSKP1-like* genes with single position-conserved intron as well as *OSK20* (*TaSKP1_7, TaSKP1_10, TaSKP1_1, TaSKP1_2, TaSKP1_12* and *TaSKP1_13*) (Figure 2).

Kong et al. [22] suggested that ancestral genes such as *ASK1* and *OSK1* are essential and as such, they evolve slowly and display wide and high expression levels. On the other hand, moderately and rapidly evolving members are more diverse in their genomic structure and are generally less expressed, suggesting that they may have lost their original function(s) and/or acquired new function(s). To test this hypothesis, we further explored the level of expression of *TaSKP1-like* genes in different tissues (grains, spikes, leaves and roots) during wheat development.

### 2.7. Expression Profiles of the TaSKP1-Like Genes during Wheat Development

To study how *TaSKP1-like* genes behave during development and in response to stimuli, we selected two RNAseq datasets from the www.wheat-expression.com platform (see Materials and Methods). After filtering the data to eliminate the genes that had a too low level of expression and the ones corresponding to "non-expressed" genes, 78 *TaSKP1-like* sequences were retained for further analysis. Comparison of normalized levels of expression during different stages of wheat development showed that *TaSKP1-like* genes exhibit variable behavior (Figure 5, Appendix A). One group of *TaSKP1-like* genes had a constitutive expression in the four analyzed tissues (grains, spikes, leaves and roots), with medium to high expression. These genes are *TaSKP1_3, TaSKP1_7, TaSKP1_10, TaSKP1_15, TaSKP1_16, TaSKP1_19, TaSKP1_21, TaSKP1_27, TaSKP1_28, TaSKP1_29, TaSKP1_30, TaSKP1_54, TaSKP1_57, TaSKP1_63, TaSKP1_82, TaSKP1_90, TaSKP1_91* and *TaSKP1_92*. However, among these genes, some had a significantly higher level of expression than others, regardless of the tissue, such as *TaSKP1_15, TaSKP1_16* and *TaSKP1_19* genes (average expression level of 11.96 ± 0.18), followed by *TaSKP1_3, TaSKP1_21, TaSKP1_27, TaSKP1_82, TaSKP1_90, TaSKP1_91* and *TaSKP1_92* genes (average expression level of 8.49 ± 0.52).

In addition, 11 other *TaSKP1-like* genes displayed a high expression level only in grains whereas they were weakly expressed in the other tissues (spike, roots and leaves) and *TaSKP1_9* displaying the highest expression level in grains.

### 2.8. Expression Profiles of the TaSKP1-like Genes in Response to Different Abiotic Stresses in Leaves

To assess the effects of abiotic stresses on the *TaSKP1-like* gene expression pattern, the data from Liu et al. [32], which include high temperature, drought and a combination of the two stresses, were extracted to identify differentially expressed genes. Using LIMMA analysis [30], we identified 16 *TaSKP1-like* genes differentially expressed in at least one treatment (Figure 6a), some were activated under stress while others were repressed. In particular, *TaSKP1_64* and *TaSKP1_65* were found to be up-regulated by heat or a combination of drought and heat one hour after treatment (Figure 6b). This result suggests that there is some degree of functional specialization within the members of this family. When analyzing the correlation between expressions of these genes, it appeared that some of them had strongly positively correlated expression profiles (Pearson coefficient (Pc) close to 1). For example, this was the case for *TaSKP_10* and *TaSKP_14* (Pc = 0.98) or *TaSKP_72* and *TaSKP_74* genes (Pc = 0.99). On the other hand, other genes had a weak expression profile, such as *TaSKP_71* and *TaSKP_72* (Pc = 0.11) or *TaSKP_14* and *TaSKP_54* genes (Pc = 0.12). It is interesting to note that some *TaSKP1-like* genes had entirely opposite profiles (Pc close to −1) such as *TaSKP_36* and *TaSKP_82* (Pc = −0.96) or *TaSKP_63* and *TaSKP_65* genes (Pc = −0.95). These results suggest that there is a strong redundancy between some *TaSKP1-like* genes, but also that there are strong divergences that could be related to some specialized functions (Figure 7).

### 2.9. Yeast Two-Hybrid (Y2H) Interaction between TaSKP1, FBX Proteins and Western Blots

An experiment was designed to test interactions between three TaSKP1 proteins and three potential FBX interactor proteins to form SCF complexes. For each TaSKP1-FBX pair, the interaction was tested in both directions (Figure 8).

The three *TaSKP1-like* genes that were selected for this study contained a single position-conserved intron. Two of these, *TaSKP1_3* and *TaSKP1_19* belong to a group of six *TaSKP1-like* genes clustered phylogenetically with six other ancestral monocot *SKP1-like* genes (*OSK1, ZmSKP1, SbSKP1, SbSKP2, BdSKP1* and *BdSKP2*) (Figure 2). The third, *TaSKP1_10* belongs to the second group of six *TaSKP1-like* genes clustered phylogenetically with three other ancestral monocot *SKP1-like* genes (*OSK20, ZmSKP2* and *SbSKP3*).

*TaSKP1_3* and *TaSKP1_10* were moderately expressed in the four tissues analyzed (grains, spikes, leaves and roots). *TaSKP1_19* was widely and highly expressed in these same tissues (Appendix A).

Of the three TaSKP1 proteins examined, TaSKP1_3 was found to interact with all the FBX proteins tested (Figure 8b). TaSKP1_19 was found to interact with two of the three FBX proteins tested (Figure 8b). By contrast, TaSKP1_10 did not display any detectable interaction under our experimental conditions, even after 24 h of incubation at 37 °C for X-gal assays (Figure 8a,b). In order to verify that the negative interactions were not merely due to a lack of protein production by the yeast, we examined the presence of the three TaSKP1 proteins fused to the Gal4 DNA-binding domain in yeast by western-blotting with the Gal4 monoclonal antibody. We observed accumulation of fusion proteins in TaSKP1_3, TaSKP1_10 and TaSKP1_19 (Figure 8c). Therefore, we considered that the negative interactions observed with TaSKP1_10 reflected the genuine behavior of this protein in yeast two-hybrid (Y2H) assay rather than a technical artefact due to the absence of interactors.

## 3. Discussion

### 3.1. Expansion and Genomic Distribution of the TaSKP1-Like Gene Family

The number of *SKP1* genes per haploid genome varies according to the species. It ranges from one copy in humans to several dozens in certain plant species [22]. Kong et al. [26] described 31 genes in rice, 21 genes in *Arabidopsis thaliana* and Elzanati et al. [24] reported four genes in the moss *Physcomitrella patens*. In this study, we identified 92 *SKP1-like* genes in bread wheat (*Triticum aestivum*). This relatively large number could be explained partly by the hexaploid nature of wheat. Indeed, our results showed that there are 25 *SKP1-like* genes in subgenome A, 30 in B and 37 in subgenome D. These values are comparable to those found in other diploid cereals that are botanically close to wheat, such as *Oryza sativa* ssp. *indica* (31 genes) or *Oryza sativa* ssp. *japonica* (26 genes). Nevertheless, when we used the criteria suggested by the International Wheat Genome Sequencing Consortium (IWGSC) (2014) to identify homeologs (i.e., two sequences are declared as homeologs when they share at least 90% amino-acid identity over the total length of the proteins), we found eight triplets and 11 pairs yielding a total of 46 proteins. This indicates that polyploidy accounts for about half of the total number of *TaSKP1-like* genes, while the rest of the *TaSKP1-like* members have likely arisen as the result of other duplication events such as small-scale duplications and/or retroposition. 

Regarding the distribution of the *TaSKP1-like* genes according to their structure, it appears that 40 out of the 92 genes (43%) do not carry any intron (Class Ib). Similarly, 11 of the 24 *SKP1-like* genes in *Fragaria vesca* (46%) and 18 of the 38 in *Medicago truncatula* (47%) are also intronless. However, the proportion of this class of genes can be even higher in, for example, *Oryza sativa* ssp. *Indica* where they represented 73% (19 out of the 26 genes) and in *Arabidopsis lyrata* where they represented 75% (15 out of the 20 genes) [23]. Kong et al. [26] suggested that *SKP1-like* genes lacking introns are likely to be retroposon-mediated rearrangements. This has been demonstrated for at least one *SKP1-like* gene, *OSK3* in rice, for which TSD (target site duplication) signatures and a polyA stretch have been found in the sequence which constitutes vestiges of a retrotransposition event. In the case of bread wheat, we scrutinized the intronless sequences for TSD and polyA signatures, but we have not been able to detect any. This is probably due to the rapid evolution of this class of genes (fast evolving genes) as suggested by Kong et al. [26] in rice and *Arabidopsis* [22]. The probable ancient retrotransposition accompanied by successive mutations could have modified these signatures and made them non-identifiable. Alternatively, intronless genes could result from the repair of a double strand DNA break (DSB) by non-homologous end joining (NHEJ) as was suggested by Farlow et al. [33]. 

In addition to the retrotransposition phenomena that contributed to the expansion of *TaSKP1-like* genes, other types of duplications were also probably involved. Indeed, when we compared large regions containing some *TaSKP1-like* genes (Figure 4), we found that they consisted of highly duplicated small regions (small-scale duplications) that occur between chromosomes of the same subgenome (for example between 4D and 7D) as well as between non-homoeologous regions (for example between 2D and 3B). This can be explained by the hypothesis proposed by Wicker et al. [34] according to which a double strand break (DSB) could be created upon the insertion of transposable elements (TE) and that the cellular repair machinery would use a partially homologous template from another region (ectopic recombination). If the template region contains a gene, then this gene would be duplicated. Consistent with this hypothesis, *TaSKP1-like* genes are located in regions enriched with repeated sequences and TEs which may favor a DSB [31]. This hypothesis was supported by Glover et al. [35] who found regions enriched with repeated sequences and TEs in the 3B chromosome in bread wheat. Twenty-seven percent of filtered genes appeared to be non-syntenic, i.e., in non-conserved chromosomal location with other cereals; whereas only 8% of filtered genes seem to be non-syntenic in maize. These percentages suggest that the *Triticeae* lineage presents a particularly high percentage of interchromosomal duplications which is normally not a feature of large genomes. The same results in wheat were observed by Guérin et al. [36] on the expansion of the NAC (NAM (for No Apical Meristem), ATAF1 and −2, and CUC2 (for cup-shaped cotyledon)) transcription factors.

Although we have not specifically addressed the mechanisms of gain and loss of introns in the *TaSKP1-like* gene family, we did identify 18 members belonging to the class of *TaSKP1-like* genes having a single intron but at various positions. When these genes were aligned with the nearest ancestral *TaSKP1-like* genes, deletions and/or insertions in the second exon, leading to its size modification were detected. For example, the *TaSKP1_35* gene has undergone a deletion of 15 bases in the second exon compared to its ancestors. The same situation was found for several other *TaSKP1 like* genes (deletion of 39 bases in the second exon of *TaSKP1_31* and *TaSKP1_32*, two insertions of three and six bases, i.e., nine bases in total in *TaSKP_36*; or deletion of three and then 46 bases, i.e., 49 bases in exon 2 of *TaSKP_37* and *TaSKP_38*. These examples show that in addition to large genomic variations, short insertions/deletions can also lead to structural and probably functional diversity in this family. Schulman et al. [37] and Zheng et al. [38] have shown that the second exon contains 21 out of the 26 amino acids essential for the interaction between SKP1 and FBX proteins in humans and those amino acids are conserved in ancestral *OsSKP1-like* genes in rice [23]. Therefore, it is reasonable to suggest that substantial modifications affecting this second exon would be likely to significantly affect the interaction profile of these proteins with FBX proteins.

### 3.2. Fate of Duplicated Genes

Extensive studies in animal and plants have shown the prominent role of duplication in creating variations within genes which may lead to adaptive responses of species to the environment. Some of the adaptive responses could be attributed to differences in expression between the duplicated genes [39]. When we compared the expression profiles of different *TaSKP1-like* genes, we found that they behave differently and that they could be separated into three categories: weakly, moderately and strongly expressed genes. In addition, even if some genes are phylogenetically very similar, they can display very divergent or even opposite expression profiles. This is the case, for example, for the *TaSKP_71* and *TaSKP_72* genes (Pearson coefficient = 0.11) or *TaSKP_63* and *TaSKP_65* (Pearson coefficient = -0.95). This observation is in agreement with the duplication-divergence-complementation (DDC) model proposed by Force et al. [40], which suggests that both duplicates could be maintained within a genome as a consequence of the accumulation of degenerative mutations in the regulatory regions of one duplicate leading to its subfunctionalization.

### 3.3. Protein–Protein Interactions

In addition to the analyses of the divergence in the expression profiles, we investigated the interaction abilities of some *TaSKP1-like* genes. It has been shown that *Arabidopsis SKP1* genes possess different interaction capabilities when tested with different FBX proteins which are the direct interactors to form specific SCF E3 ligase complexes [25,41]. Similarly, it has been shown that not all SKP proteins are able to interact with all FBX in rice [23], nor in the moss *Physcomitrella patens* [24]; but ancestral SKP1 proteins interact with a large number of FBX proteins. A similar situation was observed in *Arabidopsis thaliana* with ASK1, ASK2 and ASK11 proteins [25,41]. In wheat, using the yeast two-hybrid (Y2H) method, the three TaSKP proteins tested were found to interact differently with the three FBX proteins tested. The TaSKP1_3 protein was able to interact with the three FBX that we tested, TaSKP1_19 with two of the FBX proteins and, finally, TaSKP1_10 did not interact with any of the FBX proteins tested. This clearly showed that these three TaSKP proteins have also diverged in their ability to interact with their FBX partners. In addition, it is interesting to note that TaSKP1_3 and TaSKP1_19 belong to an ancestral monocot group (OSK1, ZmSKP1, SbSKP1, SbSKP2, BdSKP1 and BdSKP2) (Figure 2). Therefore, we hypothesize that six TaSKP1 proteins (encoding TaSKP1_3, TaSKP1_15, TaSKP1_16, TaSKP1_19, TaSKP1_21, TaSKP1_27) are likely to have functions similar to ASK1 and ASK2 in *Arabidopsis*, OSK1 and OSK20 in rice and PpSKP1 and PpSKP2 in *P. patens*.

## 4. Materials and Methods 

### 4.1. Data Retrieval

The annotated proteins in the database of the International Wheat Genome Sequencing Consortium (IWGSC RefSeq v1.0) and in the TGAC v1 [28] were searched using the HMMER2 program implemented in Unipro UGENE 1.25 [42]. As a query, the HMM (Hidden Markov Model) profile of SKP1 (PF01466) and SKP1_POZ (PF03931) domains were downloaded from Pfam. The HMM profiles were generated by alignments of 243 and 13 seed sequences (Pfam database). The HMMER2 selected proteins (with an E-value cut-off <0.1) were then scanned for SKP1 and SKP1_POZ domains using InterProScan [43] (http://www.ebi.ac.uk/interpro/search/sequence-search). The SKP1 and SKP1_POZ domains containing proteins identified by InterProScan were then used as the query sequence for a BLASTP search (with an E-value cut-off <1 × 10^–5^) of the entire wheat genome. Finally, the BLASTP hits were scanned for SKP1 and SKP1_POZ domains using InterProScan. The end results of the combination of the HMMER2, BLASTP and InterProScan searches provided the whole set of TaSKP1 proteins in the two annotated references of the wheat genome.

### 4.2. Chromosomal Location of TaSKP1-Like Genes

To map the *TaSKP1-like* genes on the *T. aestivum* L. pseudomolecule, the chromosomal coordinates of each *TaSKP1-like* gene were obtained from the International Wheat Genome Sequencing Consortium (https://www.wheatgenome.org/) and visualized using MapGene2Chromosome V2 (http://mg2c.iask.in/mg2c_v2.0/).

### 4.3. Identification of Homoeologous Relationships between TaSKP1 Sequences

Because bread wheat is a hexaploid species, we aimed first at identifying the homoeologous genes. For this purpose, we applied the criterion proposed by Pfeifer et al. [44] and Liu et al. [32]. The CD-HIT suite (http://weizhong-lab.ucsd.edu/cdhit-web-server/cgi-bin) was used to identify clusters of highly similar sequences with the identity cut-off set at 90% over the longest sequence. The rest of CD-HIT parameters were set as default. Highly similar sequences located on similar positions of homoeologous chromosomes were deemed homoeologs. Schematic representation of the intron-exon organization of *TaSKP1-like* genes was inferred using Gene Structure Display Server 2.0 (http://gsds.cbi.pku.edu.cn) 

### 4.4. Duplications

When browsing the wheat pseudomolecule, we found that *TaSKP1-like* genes are often clustered in chromosomal areas without the presence of other genes, presumably due to duplications. In order to identify the duplication events between loci, for each locus containing a *TaSKP1-like* gene, 100 kb on either side of the gene were retrieved. In cases where multiple *TaSKP1-like* genes were separated by less than 100 kb, the window of sequence retrieval was extended from the most distal *SKP1* coordinates. Homologies between these sequences were then searched using Yass, a Blast-like tool [45] with an e-value threshold <1 × 10^-100^. The Yass output results were then parsed using the following criterion: sequences with at least 90% identity over at least 3 kb were considered duplicate. Figures showing these duplications were generated using circa software (OMGenomics, http://omgenomics.com/circa/).

### 4.5. Phylogeny of the TaSKP1-Like Genes

MEGA7 software was used to infer the unrooted phylogenetic tree, based on the alignment of the amino acid sequences of 174 *SKP-like* genes from moss, monocot and eudicot species comprising 92 *T. aestivum TaSKP1-like* genes (*TaSKP1_1* through *TaSKP1_92*), four *P. patens* genes denoted *PpSKP* (*PpSKP1* through *PpSKP4*), 21 *A. thaliana ASK* genes (*ASK1* through *ASK21*) and 29 *Oryza sativa* genes (*OSK1* through *OSK31*; two of which are no longer included in this denomination). A set of 28 additional *SKP1-like* genes from various plant species with a single position-conserved intron was included in the phylogeny tree. The conservation of intron position was used as a criterion to identify putative ancestral *TaSKP1-like* genes, as suggested by Kong et al. [26] and Kahloul et al. [23]. The position of a given intron was considered as conserved if it occurred between two aligning bases in the alignment of the coding sequences [46].

Evolutionary history was inferred using the maximum likelihood method [47]. The parameters of the constructed tree were: test of phylogeny: (a) bootstrap (500 replicates), (b) gaps/missing data treatment: partial deletion, (c) model/method LG model, (d) rates among sites: gamma distributed with invariant sites (G). Only bootstrap values greater than 60 were displayed on the tree. 

### 4.6. Meta-Analysis of the Expression of Wheat SKP1-Like Genes

The expression profiles of *TaSKP1-like* genes were monitored in different tissues (grains, spikes, leaves and roots, data extracted from Choulet et al. [29]) or in response to abiotic stresses in leaves (heat shock and PEG-induced drought at an early stage, data extracted from Liu et al. [32]), based on the data available on the www.wheat-expression.com platform [27]. Briefly, raw expression data were downloaded and analyzed using the R Package LIMMA [48]. Weakly expressed genes were removed from the whole wheat genome expression raw data. The expression levels of the remaining genes were normalized using the TMM and Voom methods [31,48]. Genes that display an absolute log fold change >1 with an adjusted *p*-value < 0.05 between two conditions were deemed Differentially Expressed Genes and noted as DEG. Hierarchical clustering of DEG was conducted using Mev software [49] with euclidean distance and average linkage.

### 4.7. Plant Materials and Growth Conditions

Recital, a winter-wheat (*Triticum aestivum*) variety, was used in this study. Plants were grown in soilless compost with 16 h light and 8 h darkness at 18 ± 1 °C. Light intensity was approximately 200 µE/m²/sec.

### 4.8. Total RNA Extraction and Isolation of TaSKP and TaFBX cDNAs

Total RNA was isolated from two to three week old wheat plants using the method described by Bogorad et al. [50]. RNA was treated with DNaseI (Invitrogen, Carlsbad, CA, USA). First strand cDNA was synthesized using the SuperScript II reverse transcriptase and oligo-dT primer (Invitrogen). Forward and reverse primers containing AttB1 and AttB2 tails at their 5’-end (Appendix A) were used to amplify *TaSKP1_3*, *TaSKP1_10*, *TaSKP1_19* and three *TaFBX* full length cDNA (Gateway Technology, Invitrogen, Carlsbad, CA, USA). PCR products were then separated on a 1% agarose gel and purified with GFX Purification Kit (Amersham, Little Chalfont, UK).

### 4.9. cDNA Cloning

To clone genes of interest, we used pENTR™/D-TOPO® cloning kits (Invitrogen, Carlsbad, CA, USA). PCR products were cloned directionally by adding four bases to the forward primer (CACC). All constructs were sequenced.

### 4.10. Binary Yeast Two-Hybrid Analysis (Y2H)

#### 4.10.1. Plasmid Constructs

To produce hybrid proteins, Gateway Cloning Technology (Invitrogen) was used. The yeast expression vectors pDEST™32 and pDEST™22 were used to generate GAL 4 DNA Binding Domain (GAL4BD) and GAL4 DNA Activation Domain (GAL4AD) fusion proteins. The LR reaction (Gateway Technology, Invitrogen) was performed in order to clone three *TaSKP1-like* genes (*TaSKP1_3, TaSKP1_19* and *TaSKP1_10*) and three *TaFBX* (*TaZTL-like*, *TaFBP7-like* and *TaABA-T-like*) in both directions into the destination vectors.

#### 4.10.2. Yeast Two-Hybrid (Y2H) Screening and Assays

Bait and prey vectors were mixed and transformed into the MaV203 yeast strain using the ProQuestTM Two-hybrid System (Invitrogen), according to the manufacturer’s manual. Y2H screens and assays were performed as described previously by Kahloul et al. [23].

### 4.11. Protein Extraction and Immunoblotting

In order to verify for the real production of fusion proteins, total yeast proteins were extracted according to the method described by Printen and Sprague [51] and separated by electrophoresis with the Mini-PROTEAN TGX (Tris-Glycine eXtended, BioRad, Hercules, CA, USA) on either precast 10% gels or 8% SDS-polyacrylamide gels. Immunoblot assay was performed as described previously by Kahloul et al. [23].

## 5. Conclusions

In this study, we identified 92 *TaSKP1-like* genes using the latest version of the wheat genome annotation. Those genes are located in regions rich in short duplications, suggesting that they have probably been expanded through small-scale duplications. Following these duplications, the *TaSKP1-like* genes have, in all likelihood, diverged by accumulating mutations to the point of exhibiting different expression profiles but also different interaction capabilities with their potential FBX protein interactors. Structural, phylogeny, expression and interaction data suggest that a group of six *TaSKP1-like* genes (*TaSKP1_3, TaSKP1_15, TaSKP1_16, TaSKP1_19, TaSKP1_21, TaSKP1_27*) which clustered phylogenetically with six other ancestral monocot *SKP1-like* genes (*OSK1, ZmSKP1, SbSKP1, SbSKP2, BdSKP1* and *BdSKP2*) could be proposed as functionally equivalent to ancestral *SKP1* genes in plants. 

## Figures and Tables

**Figure 1 ijms-20-03295-f001:**
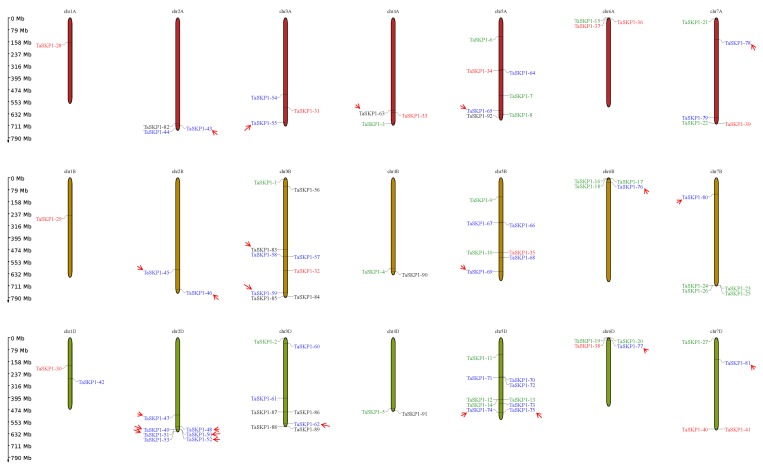
Chromosomal repartition of the 92 *TaSKP1-like* genes on the 21 chromosomes of the wheat genome (A, B and D subgenomes). Colors indicate genes belonging to the same structural class (see Appendix A). Red arrows indicate the genes considered for duplication analysis (see Appendix A).

**Figure 2 ijms-20-03295-f002:**
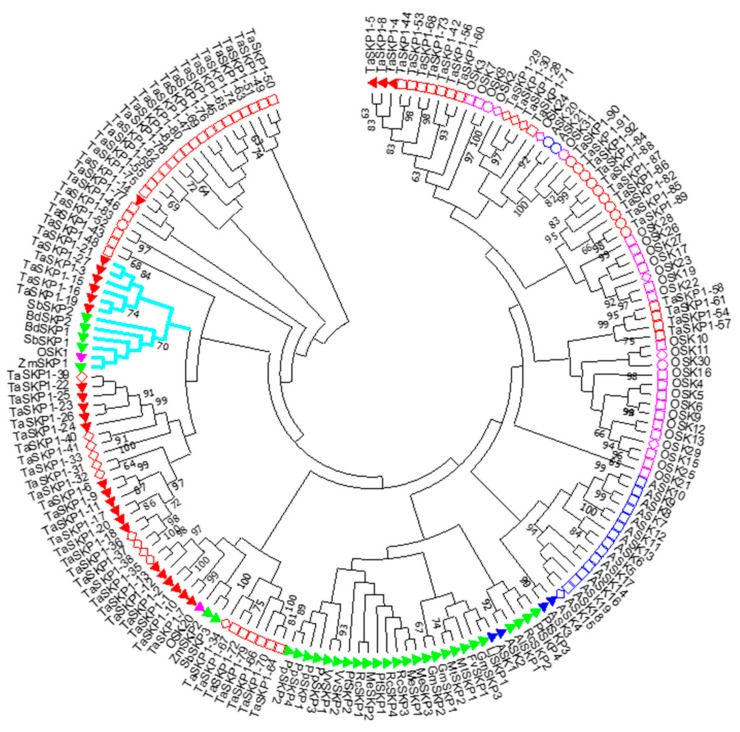
Phylogenetic tree of *SKP1-like* genes in wheat based on the alignment of the amino acid sequences of 174 *SKP-like* genes from moss, monocot and eudicot species comprising 92 *T. aestivum TaSKP1-like* genes (*TaSKP1_1* through *TaSKP1_92* in red), 4 *P. patens* genes denoted *PpSKP* (*PpSKP1* through *PpSKP4* in green), 21 *Arabidopsis thaliana ASK* genes (*ASK1* through *ASK21* in blue), 29 *Oryza sativa* genes (*OSK1* through *OSK31* in purple; two of which are no longer included in this denomination). A set of 28 additional *SKP1-like* genes from various plant species with a single position-conserved intron was included (*ZmSKP* from *Zea mays*, *SbSKP* from *Sorghum. bicolor*, *MtSKP* from *Medicago truncatula*, *RcSKP* from *Ricinus communis*, *PtSKP* from *Populus trichocarpa*, *GmSKP* from *Glycin max*, *VvSKP* from *Vitis vinefera*, *BdSKP* from *Brachypodium distachyon*, *LjSKP* from *Lotus japonicus*, *AlSKP* from *Arabidopsis lyrata*, *MeSKP* from *Medicago. esculenta*). The clade colored in blue contains the genes with a single intron in a conserved position.

**Figure 3 ijms-20-03295-f003:**
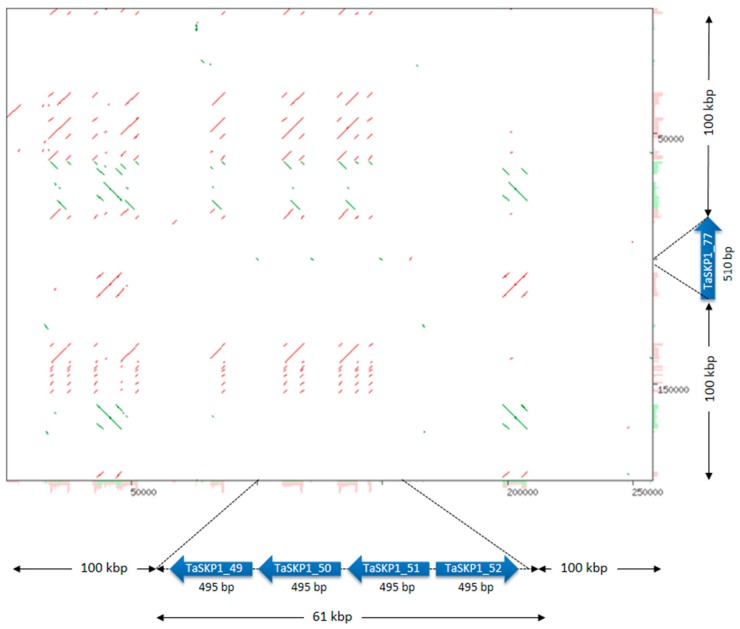
Dotplot showing an example of duplications between two genomic regions containing *TaSKP1_49, TaSKP1_50, TaSKP1_51* and *TaSKP1_52* genes and *TaSKP1_74* and *TaSKP1_75* genes. The red dots correspond to the sense duplications and the green dots to the antisense duplications.

**Figure 4 ijms-20-03295-f004:**
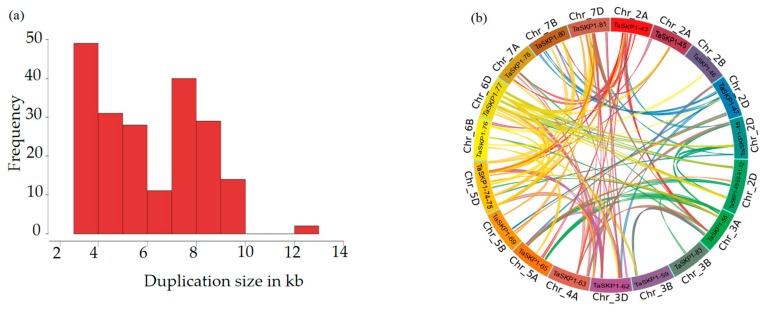
(**a**) Frequency and size of duplicated regions on the wheat genome and (**b**) duplicate events in the relevant phylogenetic clades. Each connection represents a duplicated sequence with a minimum size of 3 Kb and at least 90% nucleic identity.

**Figure 5 ijms-20-03295-f005:**
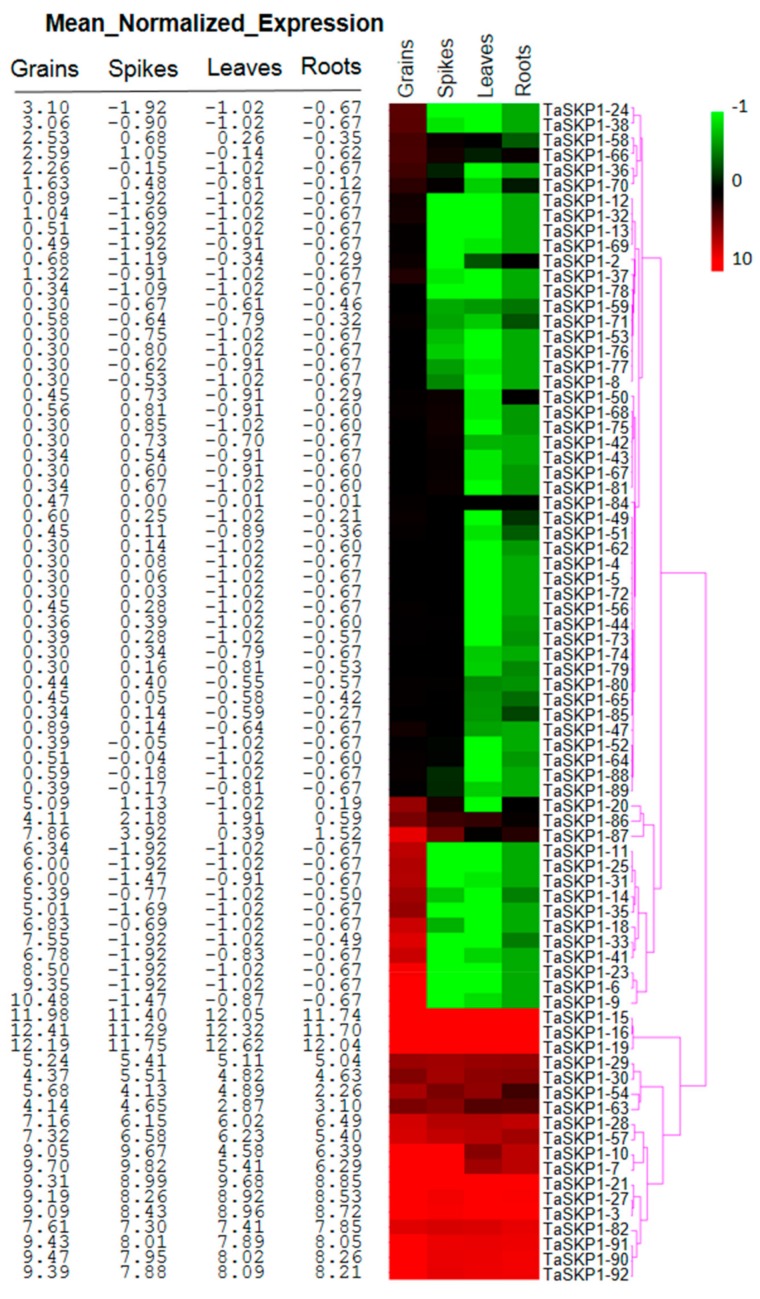
Heat map of the 92 *TaSKP1-like* genes’ expression profiles in wheat grains, spikes, leaves and roots. Hierarchical clustering was performed for the transcript abundance from all tissues. The values indicate the expression level of each gene normalized using the TMM and Voom methods [30,31] after removing non-expressed genes. Highly expressed genes are highlighted in red, weakly expressed genes are highlighted in green.

**Figure 6 ijms-20-03295-f006:**
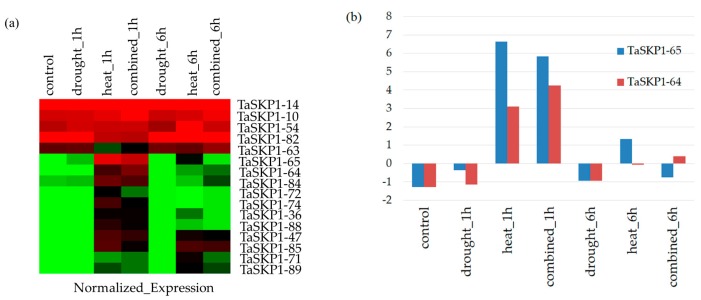
(**a**) Heatmap of 16 *TaSKP1-like* genes differentially expressed under 1 h of drought, heat and a combination of drought and heat from Liu et al. [32] and (**b**) expression profiles of two *TaSKP1-like* genes presenting a high expression under heat and a combination of heat and drought stresses.

**Figure 7 ijms-20-03295-f007:**
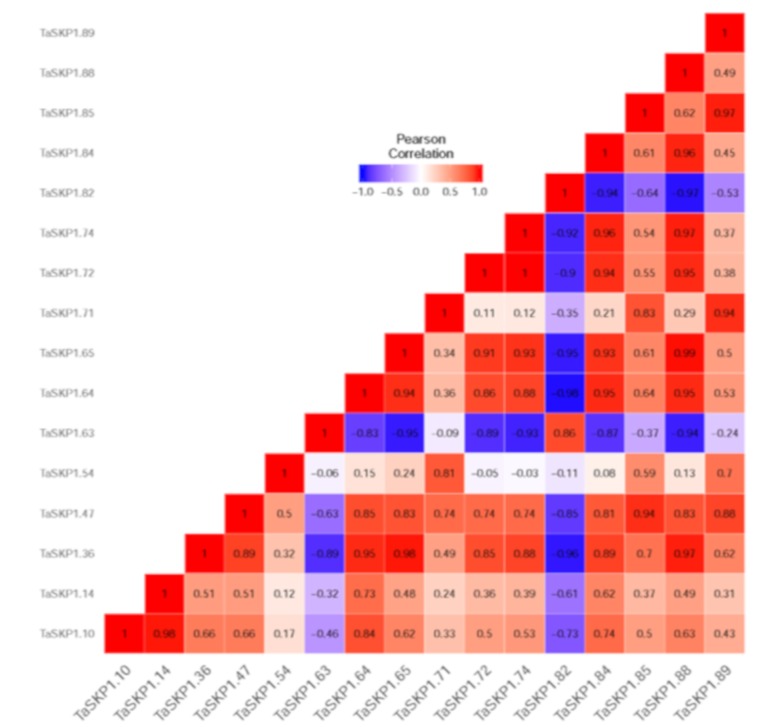
Pearson correlations between 16 *TaSKP1* differentially expressed genes under 1 h of drought, heat and a combination of drought and heat from Liu et al. [32]. Dark red color indicates a strong positive correlation. Dark blue color indicates a strong negative correlation.

**Figure 8 ijms-20-03295-f008:**
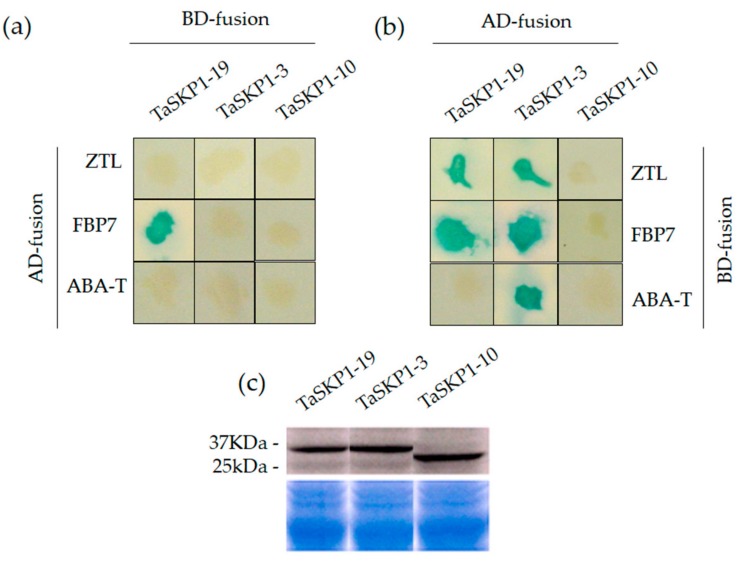
(**a**) and (**b**) Interactions of three TaSKP1 (TaSKP1-19, -3 and -10) proteins with three FBX proteins (ZeitLoop: ZTL, ABA-Tubby: ABA-T, and FBX7: FBP7) determined using yeast two-hybrid (Y2H) analysis. To generate prey (AD-fusion) and bait (BD-fusion) constructs, we fused full-length TaSKP1 and FBX proteins with the GAL4 activation and GAL4 DNA binding domains, respectively. The constructs were tested with both combinations through X-gal assays showing a blue color for positive interactions whereas non-interacting proteins remained white. (**c**) Immunoblot analysis of TaSKP1 and FBX proteins fused to the Gal4 DNA-binding domain.

**Table 1 ijms-20-03295-t001:** The distribution of 92 *TaSKP1-like* genes (International Wheat Genome Sequencing Consortium (IWGSC) RefSeq) in *T. aestivum* 21 chromosomes.

Chromosome	A Subgenome	B Subgenome	D Subgenome
1	1	1	2
2	3	2	7
3	3	9	8
4	3	2	2
5	7	7	10
6	3	4	4
7	5	5	4

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
