# Peer review of "Expansion and Functional Diversification of SKP1-Like Genes in Wheat (Triticum aestivum L.)"

_ijms, 2019, doi:10.3390/ijms20133295_

Reviewer 1 Report

Figures' quality was too low to understand anything, and due to that, I was unable to understand how they tied the text together.  Language can be improved to make it easily comprehensible for the readers.  Major headings seem to be out of order.  

Author Response

- We apology for the lack of readability of the figures. We transformed the .jpeg figures into .tiff that are correct now.

- The manuscript was read and edited by an English-native professional.

- We are sorry, but we double-checked the headings and in our opinion, they seem accurate.

Reviewer 2 Report

Selective degradation of proteins by E3 ubiquitin ligases plays a key role in diverse aspects of eukaryotic mechanisms. Among the multi-subunit E3 ubiquitin ligases complexes, the SKP1-CUL1-F36 box protein (SCF) family is particularly well characterized. Each SCF complex is composed of four protein components: the Cul1-Rbx1 catalytic core bound to a variable F-box protein (FBX)-Skp1 substrate recognition module. SKP1 is a small protein of approximately 160 amino acids and functions as a core component connecting the CUL1 and an F-box protein. SKP1 plays crucial roles in cell-cycle progression, in hormone and light signaling as well as in vegetative and flower development.

In this study, all Triticum aestivum SKP1-like (TaSKP1) genes were retrieved using data from the latest release of the wheat genome database (IWGSC RefSeq v1.0) (https://www.wheatgenome.org/) and TGAC v1 annotated genome reference respectively. Analysis of the structure, the expansion and the expression of this family of genes was conducted. In addition, since few studies have demonstrated the interactions between wheat FBX proteins and SKP1-like proteins, we used yeast two-hybrid (Y2H) systems to examine the molecular interactions between certain TaSKP1 proteins, considered to be ancestral proteins, and three wheat FBX (TaFBX proteins). These observations suggested that 6 Ta-SKP1 genes are likely to be ancestral genes, having similar functions as ASK1 and ASK2 in Arabidopsis, OSK1 and OSK20 in rice and PpSKP1 and PpSKP2 in P. patens.

This study has been well done and the manuscript was well written. However, the big defect is in the analysis of data.

Bred wheat is hexaploid species. For SKP1-like genes, we have to consider about gene duplication in genome and homoeologous copy among three genomes.

The authors have to clarify which genes are duplicated gene family and which genes have homoeologus relationship. Please summarize it.

Gene structural analysis and expression analysis has to be done by at first homoeologous gene family, and then in the duplicated genes.

The readers of this paper want to know the diversification among homoeologous gene family and among duplicated gene family separately.

And discuss about gene evolution of SKP1-like genes.

If possible, in situ hybridization analysis is needed in the expression analysis.

Author Response

We agree with the suggestions made by the reviewer #2, and added a section dedicated to the identification and expression analysis of homoeologous TaSKP1-like. This part is called: “Homoeologous group identification, structure and expression profiles of TaSKP1-like genes”. We also added a paragraph as well in the Materials and methods part called “Identification of homoeologous relashionships between TaSKP1 sequences”. Nevertheless, it may be noticed that at the beginning of the discussion section in the previous version, we have discussed the expansion of TaSKP1-like gene family in wheat and suggest that polyploidy (i.e homoeologs) contributed for about half the number of the TaSKP repertoire while the remaining sequences are the result of other duplication events.

Also, it may be noticed as well that evolution of the TaSKP1-like genes was discussed throughout the manuscript, in particular in the following sections:  

3.1. Expansion and genomic distribution of the TaSKP1-like gene family, lines 353-367

3.2. Fate of duplicated genes: lines 401-421

We agree that In-situ hybridization could be useful for identification of tissue-specific expression in compartments such as those found in the seed. However, this technique is hardly amenable to high throughput screening and necessitate specific equipments that we do not have. In contrast, RNAseq data obtained from dissected tissues could provide similar reliable information.

Reviewer 3 Report

The paper identifies and categorizes the various duplication of SKP1 genes in wheat and sets the stage for future analysis to identify their function in development and response to various stresses. This is an interesting topic, and the analysis appears sound. Throughout parts of the manuscript I felt guided through the complex topics, but occasionally the reader needs more help to understand what exactly was done or to comprehend the significance of the approach. Also, the resolution of all figures was insufficient, and the figure legends could be expended to better explain the displayed data;  I made my comments directly in the pdf. 

Author Response

We modified the figures and answered the comments as recommended directly in the text.

We expanded the figures’ captions to be more explicit.

Round  2

Reviewer 1 Report

Thanks for implementing the suggested review.